# New Record of *Dendronephthya* sp. (Family: Nephtheidae) from Mediterranean Israel: Evidence for Tropicalization?

**DOI:** 10.3390/biology12091220

**Published:** 2023-09-08

**Authors:** Hagai Nativ, Ori Galili, Ricardo Almuly, Shai Einbinder, Dan Tchernov, Tali Mass

**Affiliations:** 1Department of Marine Biology, The Leon H. Charney School of Marine Sciences, University of Haifa, Haifa 3103301, Israel; hagainativ@gmail.com (H.N.); ralmuly@univ.haifa.ac.il (R.A.); shai.einbinder@gmail.com (S.E.); dtchernov@univ.haifa.ac.il (D.T.); 2Morris Kahn Marine Research Station, The Leon H. Charney School of Marine Sciences, University of Haifa, Haifa 3780400, Israel; ori_g_inal@yahoo.com

**Keywords:** Lessepsian migration, Mediterranean Sea, alien species, Mesophotic reef

## Abstract

**Simple Summary:**

When species from one place invade and settle in another, it can cause serious problems for the local environment. These invasions, known as bio-invasions, disrupt natural ecosystems and can lead to major changes. The Mediterranean Sea is especially at risk because the water conditions are changing rapidly due to climate change, which, coupled with the opening of the Suez Canal, creates an appropriate environment for species from the Red Sea to move in. In May 2023, divers found a type of soft coral called *Dendronephthya*, that is new to the Mediterranean Sea near Israel. This coral is normally found in the Indo-Pacific region and is common in the Red Sea. Using molecular and morphological analysis, we confirmed the identity of the coral. Because this coral can attach swiftly to surfaces and grow quickly, it is expected to spread rapidly and become more common throughout the Mediterranean Sea.

**Abstract:**

Bio-invasions have the potential to provoke cascade effects that can disrupt natural ecosystems and cause ecological regime shifts. The Mediterranean Sea is particularly prone to bio-invasions as the changing water conditions, evoked by climate change, are creating advantageous conditions for Lessepsian migrants from the Red Sea. Recently, in May 2023, a new alien species was documented in the Mediterranean Sea—a soft coral of the genus *Dendronephthya.* This discovery was made by divers conducting ‘Long-Term Ecological Research’ surveys, along the coast of Israel, at a depth of 42 m. Genetic and morphological testing suggest that the species identity may be *Dendronepthya hemprichi*, an Indo-Pacific coral, common in the Red Sea. According to life history traits of this species, such as accelerated attachment to available surfaces and fast growth, we expect it to rapidly expand its distribution and abundance across the Mediterranean Sea.

## 1. Introduction

Bio-invasions are some of the most deleterious and pervasive consequences of anthropogenic global change. These invasions are capable of provoking cascade effects that can disrupt natural ecosystems and cause ecological regime shifts [1]. Given suitable environmental conditions and the fragility of an ecosystem, an introduced (alien) species has the potential to become invasive, transforming into a pest within its new habitat and spreading rapidly [2].

In the marine world, bio-invasions are often exacerbated by climate change as water temperature, in particular, is regarded as a crucial factor that can act as a selective filter, ultimately governing the potential success of alien marine species [3]. There are many additional factors that influence the success of a bio-invasion, and in the case of alien sessile benthic invertebrates, reproductive strategies also play a critical role [4]. Mixed species assemblages are often generated through sexual reproduction [5], while monospecific aggregations result from asexual reproduction such as budding or fragmentation [6,7,8]. Reproduction strategies are pivotal to enabling invasive species to out-compete native species through overgrowth, smothering, or competitive exclusion.

Marine alien species can be introduced to a new environment via several pathways, all of which are associated with anthropogenic activities. Global shipping is a major vector, as many species are moved across the globe in their larval stage, within ballast waters, or attached to ship hulls as biofouling organisms [9,10]. Aquaculture and the aquarium trade are also two paramount vectors, as exotic species are brought over to new geographic regions and may be released into the wild as they become undesirable as pets (aquarium trade) or as they spill over from their rearing enclosures (aquaculture) [11,12]. Lastly, navigational canals are a main pathway for the introduction of marine invasive species [10,13,14]. One example is the Suez Canal, which is the primary pathway for the introduction of over half of the non-indigenous species in the Mediterranean Sea [15,16].

The Mediterranean Sea is an ultra-oligotrophic [17], semi-enclosed temperate sea, which exhibits a high salinity of 39 ppt [18] and a wide annual temperature range (15–30 °C) [19,20]. The Mediterranean Sea is unique in its rapidly changing ecosystems, affected by both climate change and the introduction of invasive species. More specifically, the eastern Mediterranean region is experiencing rapid warming due to climate change, and in recent decades, water temperatures have been rising at a rate of 0.35 ± 0.27 °C decade^−1^ [21,22]. These trends also result in an increase in the minimum winter temperatures in coastal waters, with minimum temperatures shifting from 16 °C to 18 °C since the 1990s [23]. Collectively, these temperature shifts may provide favorable conditions for the invasion of warm-water species [2].

Across the Mediterranean Sea, over 500 alien species originating from the Red Sea have been documented to date [24,25,26,27,28], a phenomenon known as ‘Lessepsian migration’ [16,29]. These alien species (fish, invertebrates, and algae) have arrived via the Suez Canal, which has undergone several expansion projects since it was first dredged in 1869, thereby removing the depth and salinity barriers which once hindered the crossing for various species [30]. Israel’s coast has often been documented as the “first stop” for alien species that later establish stable populations across the entire Mediterranean Sea [31,32,33]. This is due to Israel’s proximity to the Suez Canal, located just south of Israel’s border, coupled with the prevailing south to north currents that run along Israel’s coast as part of the larger counterclockwise circulation of the Mediterranean Sea [34].

Although various alien species from diverse phyla have been recorded in the Mediterranean, the only documented alien soft coral (Cnidaria: Octocorallia) to date is *Melithaea erythraea* (Ehrenberg, 1834). This soft coral is native to the Red Sea and has arrived in the Mediterranean via Lessepsian migration. *Melithaea erythraea* was first documented in the Hadera power plant (Israel) in 1999, and remained confined within that facility until 2015 when additional colonies were observed on the surrounding rocky reefs [35,36].

In May 2023, during a routine monitoring survey, as part of a ‘Long-Term Ecological Research’ (LTER) program conducted by the Morris Kahn Marine Research Station (MKMRS), researchers observed and documented the first record of the soft coral *Dendronephthya* sp. in the Mediterranean Sea.

Soft corals are found worldwide in a wide range of depths and water temperatures [37], and are the second most abundant sessile organism in many coral reefs [38]. Soft corals have an important functional role of ecosystem engineering, adding three-dimensional complexity to the reef environment, which increases diversity [39]. Several species of soft corals are indigenous to the Mediterranean, mainly from the families Pennatuloidea and Alcyoniidae [40,41].

The genus *Dendronephthya* (Cnidaria: Octocorallia: Alcyonaea: Nephtheidae) is a genus of soft corals found in the tropical waters of the Indo-Pacific Ocean. To date, species classification of the family Nephtheidae has mainly been based on morphology [42]. Recent phylogenetic analysis using a whole mitochondrial genome provides better resolution of the topology of this family [43,44]. However, further genetic information is required in order to resolve species-level phylogeny amongst genera in this family [42,45]. *Dendronephthya* spp. has been documented at a wide range of depths, and is primarily found in habitats with strong currents [37,46,47] such as vertical artificial structures or steep reefs [48]. *Dendronephthya* is an azooxanthellate genus [49] characterized by a wide range of bright colors, with eight pinnate tentacles on each polyp, a branching divaricate structure supported by a hydrostatic skeleton, and internal calcareous skeletal elements called sclerites [50,51,52]. It is a passive suspension feeder, dependent on ambient currents for the supply of food particles, mainly phytoplankton [49].

*Dendronephthya* spp. tends to rapidly populate the available substrate, often artificial surfaces, and recruitment can be observed in as little time as two days [47,48,49,51,53]. Attachment to artificial structures is advantageous as it often enables exposure to high currents, which in turn enhance the coral’s growth [48].

In addition to sexual reproduction [50], *Dendronephthya* spp. can also reproduce via clonal propagation [51] where autotomized fragments, which are negatively buoyant, settle on the outer face of a horizontal substratum [53]. Due its fast growth and reproductive strategies, *Dendronephthya* spp. often become the most abundant sessile organisms on these structures; Refs. [53,54] report that the number of *Dendronephthya* spp. colonies can increase four-fold in one year following initial recruitment to a new habitat.

In this study, we aim to characterize the soft coral observed in the Mediterranean Sea that we suspected to be *Dendronephthya* spp. The broader objective was to identify the species and understand the expansion potential of this species in the Mediterranean Sea; therefore, we examined both morphological characteristics and genetic information for the collected specimens of this coral.

## 2. Materials and Methods

### 2.1. Study Site and Sample Collection

Along the Mediterranean coast of Israel, underwater surveys have been conducted bi-annually at nine locations (Figure 1) over the past 9 years, as part of a ‘Long-Term Ecological Research’ (LTER) program led by the Morris Kahn Marine Research Station (MKMRS) (MKMRS LTER; established in 2014; https://med-lter.haifa.ac.il/index.php/en/data-base, accessed on 6 July 2023). Monitoring dives are carried out using Megalodon Closed Circuit Rebreathers (CCRs) to avoid disturbance to the marine life, increase bottom time, and increase safety. Four 25 m long transects are surveyed at each site, along which data on fish and invertebrate community composition are collected. Photo quadrats with an area of 25 cm^2^ are taken every 2 m along the transect line and analyzed on the Coralnet website [55] under the project ’Israeli monitoring program‘ to characterize the invertebrate community composition. Fish species, size, distance from transect, and number of individuals are recorded for characterization of the fish community composition.

On 18 May 2023, MKMRS LTER researchers encountered several colonies of a soft coral that were suspected as being *Dendronephthya* sp. along the rocky reef monitoring site (32.54° N, 034.85° E) located near Sdot Yam, Israel, at a depth of 42 m (Figure 1). This rocky reef site is exposed to open sea currents, and at the time of the survey, the water temperature was 19 °C. Specimens from three colonies at the natural rocky reef monitoring site were collected. In order to compare these specimens with native populations in the Red Sea, seven colonies were later collected in the Red Sea, adjacent to the Inter-University Institute (29.30° N, 34.54° E) in Eilat, Israel, under a special permit from the Israel Nature and Parks Authority. The seven specimens from the Red Sea were collected from both artificial structures and natural reefs. Four colonies were collected from a depth of 36 m, two of which were associated with an artificial structure and two of which were associated with the natural reef. An additional three colonies were collected at a depth of 12 m, and these three were associated with artificial structures. Samples from each colony were sectioned into fragments of 1 cm^2^ in size and were flash-frozen in liquid nitrogen prior to DNA extraction. The remainder of the specimens collected were utilized for further morphological examination. Specimens collected from the Mediterranean have been deposited at The Steinhardt Museum of Natural History, Tel Aviv University, Israel (Voucher number, SMNHTAU-Co.39049).

### 2.2. DNA Extraction

Genomic DNA was extracted using Qiagen Blood and Cell Culture DNA kit (Qiagen, USA#13323). For species identification, three genetic markers were PCR-amplified. The mitochondrial marker ribosomal 16S gene was amplified using primers DN1-F (5′-AGGCTACTTAAGTATAGGGG-3’) and DN1-R (5′-AACTCTCCGACAATATTACGC-3′), with PCR conditions as described in [43]. The second markers were the oxidase subunits I and II (cox1 and 2), which were generated based on the available sequences of *Dendronephthya hemprichi* (native to the Red Sea) in NCBI at the time (GU355996.1): DhCox12F AGAGTGTTCTCACCTACTTTAG and DhCox12R GTTTAGCAGAAAATGTGGGTAT. The third marker was the MutS-like protein (MSH1) gene which was generated based on the available sequences of *D. hemprichi* in NCBI at the time (GU356019.1): DhMsh1F GAGCCAAATACCTATGCAATAT and DhMsh1R ACACGGCAAGTTGGTTAGTG. All PCRs were performed using the Kodaq 2X PCR MasterMix (ABM, Richmond, BC, Canada) following the manufacturer’s protocol. DNA yield and PCR products were analyzed via electrophoresis on a 1.0% agarose TBE (90 mM TRIS-borate and 2 mM EDTA) gel run at 110 V. PCR products were Sanger-sequenced in both directions using the amplification primers on an ABI 3730xl DNA Analyzer (Applied Biosystems^TM^, Waltham, MA, USA) at HyLabs (Rehovot, Israel).

In total, 27 complete mitochondrial genome sequences of the order of Malacalcyonaceae were retrieved from the NCBI database. Table 1 lists the sequences used for analysis, including those obtained from the samples collected for this study. Sequences from each gene were (separately) aligned using the MUSCLE algorithm in MEGA11 [56]. Alignments were trimmed to retain shared regions among all sequences. The trimmed alignment of the rRNA gene MSH1 and cox1 gene included 534 bases, 661 bases, and 775 bases, respectively. Then, the alignments were concatenated and used for the calculation of a maximum likelihood tree. Only species/samples for which sequences were available for at least two genes were used. A maximum likelihood tree was calculated using the PhyML 3.0 algorithm [57] and the web application at http://www.atgc-montpellier.fr/phyml/ (accessed on 6 July 2023). Standard bootstrap analysis was performed with 1000 repeats.

### 2.3. Sclerite Morphological Analysis

Sclerites were isolated from segments of coral tissue, 1–2 cm in length, using 3% sodium hypochlorite. Once tissue was no longer visible, the remaining sclerites were rinsed three times with DDW and stored in 100% EtOH. Sclerites were placed on silica wafer, mounted on SEM plugs, and vacuum-coated with 5 nm Au/Pb (80:20%) prior to examination under a ZEISS SigmaTM SEM (ZEISS, Oberkochen, Germany), using an SE2 detector (1–2 kV, WD = 6–7 mm). The length and width of the different types of sclerites found in the polyps were measured using FIJI [58] (*n* = 151 sclerites).

## 3. Results

### 3.1. Macro Morphological Analysis

In total, fifteen colonies of soft corals, that were suspected as being *Dendronephthya* spp., were observed in the Mediterranean, in a 10 m^2^ area of rocky reef, with colony sizes ranging from 5 to 50 cm (Figure 2), suggesting the utilization of propagation strategies. This particular area of rocky reef is one of the nine sites that has been routinely monitored every year in both the spring and fall, since 2014. Routine surveys include both photo quadrats for assessments of percent cover of algae and invertebrates, as well as fish counts. At the remaining eight rocky reef monitoring stations, during the spring 2023 survey, no additional evidence of any soft corals were found. Furthermore, no soft corals were observed during any of the previous survey years (https://med-lter.haifa.ac.il/index.php/en/data-base, accessed on 6 July 2023).

The observed colonies displayed distinctive traits associated with the genus *Dendronephthya*. These included vivid red pigmentation, as well as their intricate branched divaricate structure. Furthermore, the polyps exhibited a notable arrangement of spindle-shaped sclerites in their armature, both within the polyp itself and on the surface of the stalk [37]. Notably, the polyps displayed a conspicuous supporting bundle of large red spindle-shaped sclerites (Figure 2C). Lastly, similar thick spindle-shaped sclerites were observed on the surface of the stalk (Figure 2D).

### 3.2. Molecular Analysis

To confirm the genus identity of the soft corals observed near Sdot-Yam, a phylogenetic analysis based on mitochondrial small subunit ribosomal RNA, MSH1, and cox1 gene sequences was performed. The analysis revealed that all studied colonies belong to the genus *Dendronephthya* (Figure 3).

The phylogenetic analysis revealed that the colonies collected in the Mediterranean Sea were identical to the ones collected in the Red Sea, based on the available information. Moreover, the two most abundant species in the Red Sea are *Dendronephthya sinaiensis* and *D. hemprichi*. Our analysis further indicates that all of the colonies collected in the Mediterranean Sea are significantly different from *D. sinaiensis* and undifferentiated from *D. hemprichi*. However, as genomic data are limited for this species, and further molecular information is required in order to resolve this species classification, we could not confirm the species using mitochondrial markers. Therefore, we also conducted micro-morphological analysis to gain a better distinction among species.

### 3.3. Micro-Morphological Analysis

Microscopic examination of the sclerite morphology via SEM (Figure 4) further suggests that all samples collected in the Mediterranean Sea belong to the species *D. hemprichi*. Specifically, all colonies contain spindle-shaped sclerites up to 3 mm long, which are typical of the supporting bundle (Figure 4A). Additional sclerites were observed, including shorter spindle-shaped sclerites up to 500 µm in length, typically found on the polyp head (Figure 4B). Both types of sclerites were covered with evenly scattered warts. In addition, small, flattened, irregular-shaped sclerites typical for the stalk section of the coral (less than 200 µm) were observed (Figure 4C) [37,46]. The average length-to-width ratio of the spindles was 11.39 ± 4.29 (*n* = 151 sclerites, Appendix A), which is within *D. hemprichi*’s reported range of ratios [46].

## 4. Discussion

Considering the colony morphology and sclerite structure, in particular, the polyp armature with its supporting bundle, the soft coral population observed in this study belongs to the *Dendronephtya* genera. The combined molecular analysis with the micro-morphology analysis of the sclerites further suggests that the species may be *D. hemprichi*. To the best of our knowledge, this is the first evidence of occurrence of the Indo-Pacific *D. hemprichi* within the Mediterranean Sea. 

The genus *Dendronephthya* is common in the northern Red Sea and the Indo-Pacific [37], with the two most common species in the northern Red Sea being *D. hemprichi* and *D. sinaiensis*. *D. hemprichi* is found in a wide range of depths, while *D. sinaiensis* has been found to prefer water depths greater than 18 m, with low light intensity [59]. The morphological differences which were observed in the polyps of the two species are indicative of differences in feeding niches, mainly related to prey size [46].

Although the exact method of transport cannot be determined (drifting asexual autotomized fragments, ship ballast, or other), *D. hemprichi* likely arrived from the Red Sea, via the Suez Canal, as part of the greater and well-documented Lessepsian Migration. Although its natural habitat is the Red Sea, *D. hemprichi* is mostly found on artificial vertical structures exposed to high flow regimes [48]. The *D. hemprichi* colonies observed in this study were located at a natural rocky reef site, possibly indicating that this site may experience high exposure to currents.

To date, only one instance of Lessepsian migration involving an alien soft coral species has been observed in the Mediterranean. The gorgonian species *M. erythraea* was first reported by Fine et al. [35] in 1999. Despite its reproductive strategy, which includes a substantial capability for swift establishment, the observed gradual expansion of its distribution in the region, as documented by Grossowicz et al. [36], raises the possibility that *M. erythraea* lacks competitive characteristics or exhibits limited adaptability to the new ecological conditions encountered in its new habitat.

Although *M. erythraea* was not found to be an aggressive invader, other soft corals world-wide have been recorded as such. In the south Caribbean waters of Venezuela, *Unomia stolonifera* (family: Xeniidae) has become extremely prominent since it first appeared between 2000 and 2005, and according to [60], this species features an average percent cover of 30–80% in shallow reefs. This percent cover is far greater than that of any other benthic taxa on the reefs. In Brazilian waters, the alien Scleractinia coral species *Tubastraea* spp. has dominated large, rocky surfaces and displaced native corals and zoantharians. The main advantage of *Tubastraea* spp. comes from employing minor but continuous sexual reproduction and clonal reproduction strategies [61,62,63].

In contrast to *M. erythraea,* and more similar to *U. stolonifera*, we anticipate a rapid increase in the distribution of *D. hemprichi* throughout the Mediterranean Sea. Previous studies consistently demonstrate its swift colonization of the available substrate, with it being capable of multiplying colony numbers four-fold within a single year [48,53,54]. Furthermore, in specific environments, it frequently emerges as the dominant sessile organism, indicating its competitive prowess [47]. Notably, the clonal propagation model [51], characteristic of the *Dendronephthya* spp., finds support in the observed diversity of colony sizes within the localized area of the Mediterranean rocky reef. Thus, the future trajectory of *D. hemprichi* in the Mediterranean holds great potential for displaying its rapid expansion dynamics and ecological significance within the marine ecosystem.

In a broad perspective, the intervention and removal of alien species are nearly always preferred. However, this is more easily said than done, particularly in the marine environment. As [64] emphasized, the high environmental connectivity of water enables the rapid dispersion of species across large areas, and therefore dispersion methods and the size of affected areas should be considered when considering an intervention effort. The complete removal of marine invasive species is rare and has only been achieved in a few restricted areas, aided by early detection and rapid response. Although an elaborate guide has been prepared for monitoring marine invasive species in Mediterranean Marine Protected Areas [65], most regions and countries, including Israel, lack the resources (funds, equipment, and manpower) to execute a swift and efficient intervention.

Marine ecosystems are currently facing combined effects from climate change and local human stressors, which have the potential to induce profound shifts at the levels of species, trophic dynamics, habitats, and entire ecosystems. The precise outcome of these interactions varies depending on the specific nature of the interaction itself [66]. Lessepsian migration serves as an exemplary illustration of such interplay. The construction of the Suez Canal and subsequent intensified shipping activities have facilitated the arrival of alien species, while climate change has concurrently reshaped the aquatic environment, rendering it conducive for the establishment of viable populations. In fact, some researchers have gone so far as to suggest that the combined effects of climate change and localized human stressors could potentially drive certain local species toward functional extinction, as emphasized by Edelist et al. [67].

In some studies, the processes observed in the Mediterranean Sea are referred to as ‘Tropicalization’ [68,69]. This overarching concept encapsulates the overall transition of the region from a temperate ecosystem toward one with tropical characteristics. Such a transformation holds diverse implications for the biota inhabiting these waters. On one hand, it may offer a competitive advantage to thermophilic species, enabling their proliferation. On the other hand, species with lower thermal tolerance may be compelled to seek refuge in deeper, cooler waters [70], or find themselves operating at the boundaries of their physiological limits. Additionally, there have been reports of poleward range expansions in several cnidarian species [71,72]. These findings collectively underscore the complex dynamics at play in response to the combined impacts of climate change and local human stressors within marine ecosystems.

## 5. Conclusions

Once an alien species has migrated to a new environment, its long-term survival and successful establishment as a population rely not only on favorable environmental conditions [73], but also on various additional factors encompassing phenotypic plasticity, competition dynamics, predator–prey interactions, and reproductive strategies [10]. In the case of *D. hemprichi*, this particular species demonstrates several advantageous traits that will likely contribute to its successful establishment in the Mediterranean region, as species originating in the highly biodiverse Red Sea are typically more competitive than species native to the Mediterranean Sea. Furthermore, this soft coral faces minimal predation pressure, and its main mode of propagation involves asexual reproduction [50,51]. However, due to the insufficiency of currently available genetic information, we could not conclusively identify the alien *Dendroenphthya* sp. as *D. hemprichi*. This stresses the importance of worldwide initiatives for obtaining full genome sequences for corals in general.

Amidst the doomsday predictions of a collapsing ecosystem, bio-invasions in the Mediterranean Sea can also be observed through rose-colored glasses. As the Mediterranean Sea is a remnant of the Tethys Ocean [10], from a historical perspective, the changes observed can also be viewed as a return to the sea’s tropical origins [74].

## Figures and Tables

**Figure 1 biology-12-01220-f001:**
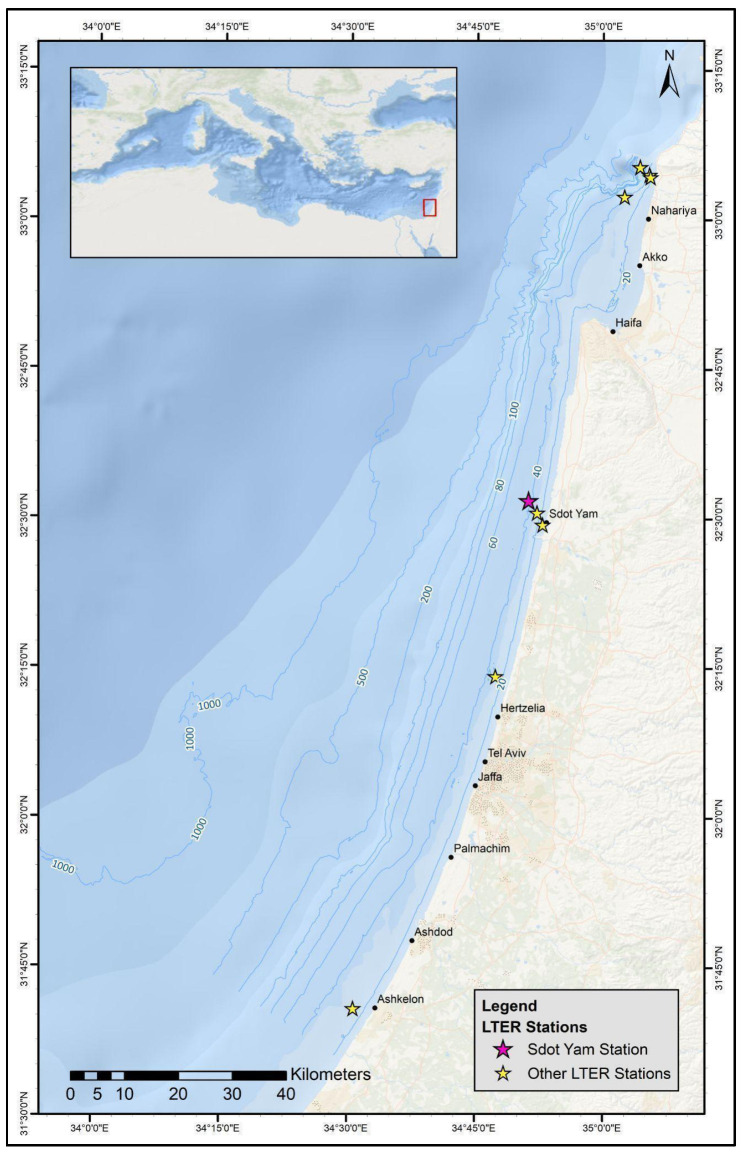
Map of the Israeli Mediterranean coast, and the nine MKMRS LTER monitoring sites.

**Figure 2 biology-12-01220-f002:**
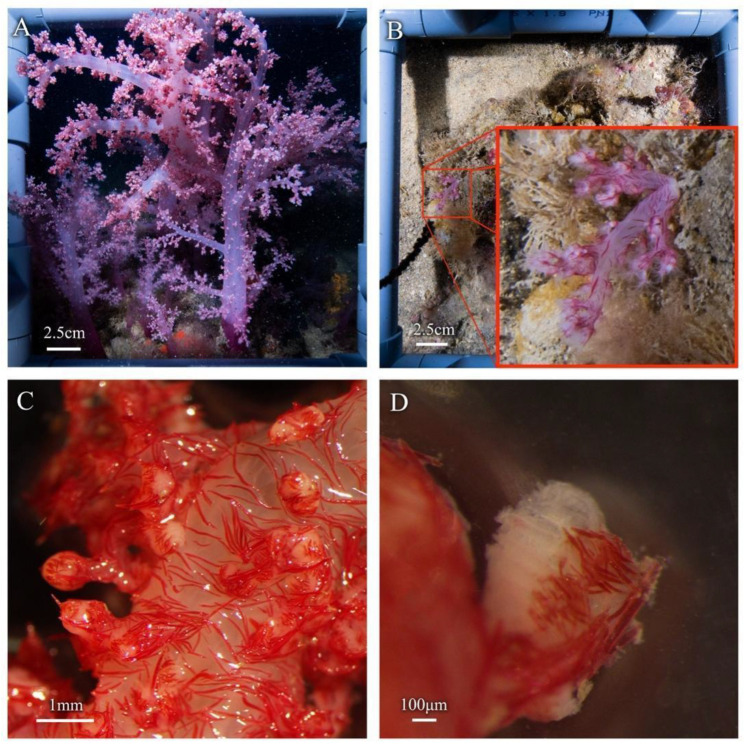
*Dendronephthya* sp. colonies observed at the rocky reef at 42 m near Sdot-Yam, Israel. (**A**,**B**) In situ images of the observed size variation in the colonies. The photographed colonies vary from (**A**) ~50 cm to (**B**) ~5 cm from base to top. (**C**,**D**) Ex situ light microscopy images of section collected from (**A**). (**C**) Red spindle-shaped sclerite arrangement on the stalk surface. (**D**) Supporting bundle and loose spindle sclerites are arranged around the polyp head.

**Figure 3 biology-12-01220-f003:**
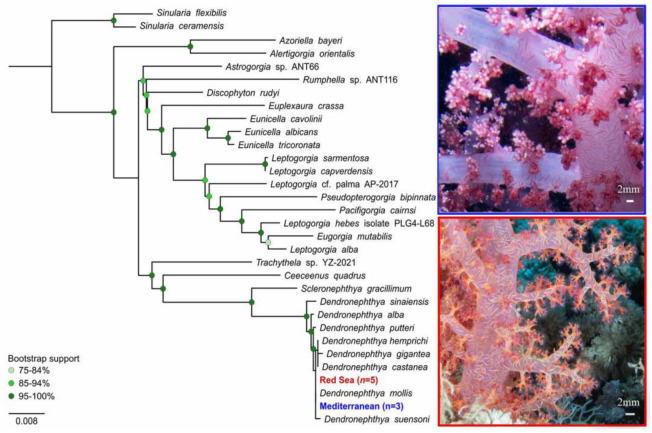
Maximum likelihood tree based on sequences of mitochondrial small subunit rRNA gene, MSH1 gene, and cox1 gene. Tree was calculated using PhyML 3.0 algorithm. Nodes denoted in green scale circles indicate bootstrap support values (percent of 1000 repeats). Scale bar represents the number of substitutions per site. Red and blue text refer to colonies analyzed in this study.

**Figure 4 biology-12-01220-f004:**
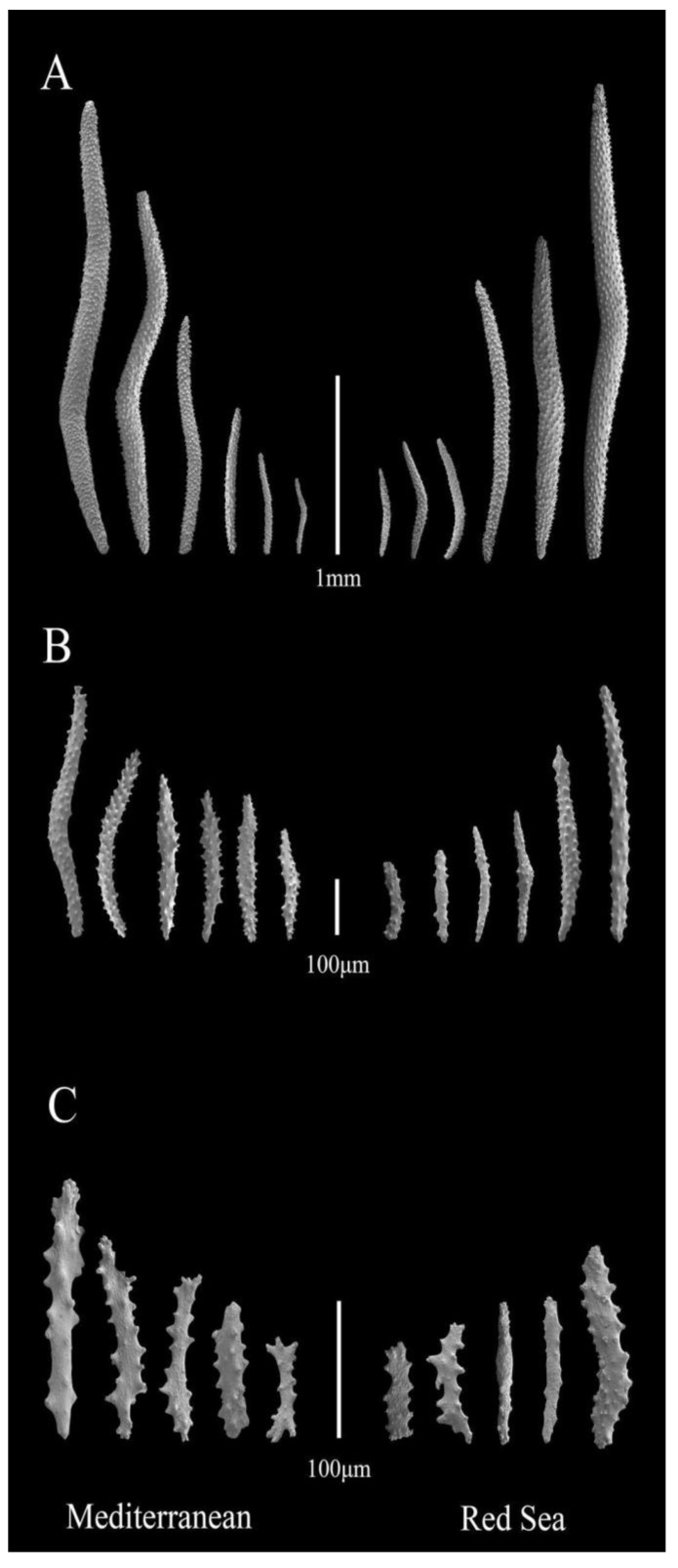
Scanning electron micrograph images of spindle-shaped sclerites from the Mediterranean Sea (**left**) and Red Sea (**right**) colonies. (**A**) Spindles from the supporting bundles. (**B**) Spindles from the polyp heads. (**C**) Irregular-shaped sclerites from the stalks.

**Table 1 biology-12-01220-t001:** Sequences used for phylogenetic analysis, including those from specimens collected in this study.

Species/Specimen	Family	Genus	Source	NCBI Accession Number	rRNA	MSH	COX1
*Dendronephthya mollis*	Nephtheidae	*Dendronephthya*	NCBI, complete genome	NC_020456.1	NC_020456.1	NC_020456.1	NC_020456.1
*Dendronephthya suensoni*	Nephtheidae	*Dendronephthya*	NCBI, complete genome	GU047878.1	GU047878.1	GU047878.1	GU047878.1
*Dendronephthya putteri*	Nephtheidae	*Dendronephthya*	NCBI, complete genome	JQ886185.1	JQ886185.1	JQ886185.1	JQ886185.1
*Dendronephthya sinaiensis*	Nephtheidae	*Dendronephthya*	NCBI, complete genome	NC_062008.1	NC_062008.1	NC_062008.1	NC_062008.1
*Dendronephthya alba*	Nephtheidae	*Dendronephthya*	NCBI, complete genome	MW423625.1	MW423625.1	MW423625.1	MW423625.1
*Dendronephthya castanea*	Nephtheidae	*Dendronephthya*	NCBI, complete genome	NC_023343.1	NC_023343.1	NC_023343.1	NC_023343.1
*Dendronephthya gigantea*	Nephtheidae	*Dendronephthya*	NCBI, complete genome	NC_013573.1	NC_013573.1	NC_013573.1	NC_013573.1
*Dendronephthya hemprichi*	Nephtheidae	*Dendronephthya*	NCBI			GU356019.1	GU355996.1
*Scleronephthya gracillimum*	Nephtheidae	*Scleronephthya*	NCBI, complete genome	NC_023344.1	NC_023344.1	NC_023344.1	NC_023344.1
*Eunicella albicans*	Eunicellidae	*Eunicella*	NCBI, complete genome	NC_035666.1	NC_035666.1	NC_035666.1	NC_035666.1
*Eunicella tricoronata*	Eunicellidae	*Eunicella*	NCBI, complete genome	NC_062012.1	NC_062012.1	NC_062012.1	NC_062012.1
*Eunicella cavolinii*	Eunicellidae	*Eunicella*	NCBI, complete genome	NC_035667.1	NC_035667.1	NC_035667.1	NC_035667.1
*Trachythela* sp. YZ-2021	Eunicellidae	*Trachyela*	NCBI, complete genome	MW238423.1	MW238423.1	MW238423.1	MW238423.1
*Eugorgia mutabilis*	Gordoniidae	*Eugorgia*	NCBI, complete genome	NC_035665.1	NC_035665.1	NC_035665.1	NC_035665.1
*Leptogorgia hebes* isolate PLG4-L68	Gordoniidae	*Leptogorgia*	NCBI, complete genome	MN052677.1	MN052677.1	MN052677.1	MN052677.1
*Leptogorgia alba*	Gordoniidae	*Leptogorgia*	NCBI, complete genome	NC_035669.1	NC_035669.1	NC_035669.1	NC_035669.1
*Leptogorgia* cf. *palma* AP-2017	Gordoniidae	*Leptogorgia*	NCBI, complete genome	KY559406.1	KY559406.1	KY559406.1	KY559406.1
*Leptogorgia capverdensis*	Gordoniidae	*Leptogorgia*	NCBI, complete genome	NC_035663.1	NC_035663.1	NC_035663.1	NC_035663.1
*Leptogorgia sarmentosa*	Gordoniidae	*Leptogorgia*	NCBI, complete genome	NC_035670.1	NC_035670.1	NC_035670.1	NC_035670.1
*Pseudopterogorgia bipinnata*	Gordoniidae	*Antillogorgia*	NCBI, complete genome	NC_008157.1	NC_008157.1	NC_008157.1	NC_008157.1
*Pacifigorgia cairnsi*	Gordoniidae	*Pacifigorgia*	NCBI, complete genome	NC_035668.1	NC_035668.1	NC_035668.1	NC_035668.1
*Astrogorgia* sp. ANT66	Astrogorgiidae	*Astrogorgia*	NCBI, complete genome	OL616212.1	OL616212.1	OL616212.1	OL616212.1
*Discophyton rudyi*	Discophytidae	*Discophyton*	NCBI, complete genome	NC_061276.1	NC_061276.1	NC_061276.1	NC_061276.1
*Alcyonium acaule*	Alcyoniidae	*Alcyonium*	NCBI, complete genome	NC_061273.1	NC_061273.1	NC_061273.1	NC_061273.1
*Alertigorgia orientalis*	Alcyoniidae	*Alertigorgia*	NCBI, complete genome	NC_061994.1	NC_061994.1	NC_061994.1	NC_061994.1
*Sinularia ceramensis*	Sinulariidae	*Sinularia*	NCBI, complete genome	NC_044122.1	NC_044122.1	NC_044122.1	NC_044122.1
*Sinularia flexibilis*	Sinulariidae	*Sinularia*	NCBI, complete genome	NC_061282.1	NC_061282.1	NC_061282.1	NC_061282.1
*Siphonogorgia godeffroyi*	Nidaliidae	*Siphonogorgia*	NCBI, complete genome	NC_062032.1	NC_062032.1	NC_062032.1	NC_062032.1
*Rumphella* sp. ANT116	Plexauridae	*Rumphella*	NCBI, complete genome	OL616268.1	OL616268.1	OL616268.1	OL616268.1
*Ceeceenus quadrus*	Paralcyoniidae	*Ceeceenus*	NCBI, complete genome	NC_062003.1	NC_062003.1	NC_062003.1	NC_062003.1
*Azoriella bayeri*	Cerveridae	*Azoriella*	NCBI, complete genome	NC_061999.1	NC_061999.1	NC_061999.1	NC_061999.1
*Euplexaura crassa*	Euplexauridae	*Euplexaura*	NCBI, complete genome	NC_020458.1	NC_020458.1	NC_020458.1	NC_020458.1
Red Sea 36 m			This study		OR458391	OR526527	OR520375
Mediterranean Sea			This study		OR462245	OR520989	OR520376

## Data Availability

All data needed to evaluate the conclusions of this paper are present in the paper.

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
