# Peer review of "New Record of Dendronephthya sp. (Family: Nephtheidae) from Mediterranean Israel: Evidence for Tropicalization?"

_biology, 2023, doi:10.3390/biology12091220_

Round 1
Reviewer 1 Report
Line 14 it makes sense that the water conditions in the Mediterranean are changing due to climate change, but it seems unlikely that water conditions in the Med are changing due to the Suez Canal. But obviously the Suez Canal makes it so many Red Sea species can enter the Med.
Line 16 Dendronephthya needs to be italicized
Line 21 cascading
Table 1 all the Dendronephthya species names need to be italicized
Line 172, 174, 201, 202, 204, 205, 210, 216, 253, 328, 329, 330, 331, 332, 373, 377, 381, 383, 393, 400, 411, 429. Dendronephthya, other genera names and the species name need to be italicized
Line 216 ration you probably mean ratio
Line 242 Erythraea is not capitalized, only the genus is capitalized
Line 243 at 1999. In 1999.
Line 246 this species is lacking
Line 256 unraveling displaying
Line 290 dooms day doomsday
Line 310 check journal policy on capitalizing the titles of papers. Many journals only have the first word of an article title capitalized, but have all words in a book title capitalized.
Line 318 Tubastraea needs to be in italics
Line 328, 329, 330, 331, 332, 373, 375, 377, 381, 383, 400, 411, 429, the genus names are properly capitalized, but species names should not be.
Line 370 this reference appears to be incomplete
Line 387 you probably meant “sessile”
Line 439 shouldn’t the disclaimer be on a separate line?
There is an invasive soft coral from the Indo-Pacific that has been discovered in the Caribbean in Venezuela, and has already expanded to become dominant in large patches. It would be worth mentioning that this could be what happens with D. hemprichi in the Mediterranean. This has been announced in several emails on “coral-list” run by NOAA. Not sure there are any published papers yet.
Any suggestions of how to deal with this? Some say that while removal early on, once a species has spread widely, removal is not practical.
Very good, just need to do some work on species names, which is minor.
Reviewer 2 Report
The paper reports another Lessepsian migrant from the Red Sea represented by soft coral Dendronephthya hemprichi (Nephtheidae). Authors use morphological (sclerite analysis) and molecular (two mitochondrial markers: 16S and cox1 & 2) to prove their identification. Paper is rather well written, however, some parts of the MS seems to be not logically placed or out of contest. There are couple of factual and many formal errors. The MS in its present state is not good for publication and needs serious work to be published.
I have serious concerns about about identification of the species reported. Up to date 250 nominal species reported in the genus Dendronephthya. It is rather complicated genus and several species are known from the Red sea. The authors are picking up the closest relative from Eilat and claiming that they have genetic and morphological proofs. However, genetics they are picking as a proof is not really working for this genus. Authors have to be aware that mitochondrial markers work rather bad to distinguish individual species in nephtheids. See e.g.
Cordeiro, R.T., Carpinelli, Á.N., Francini-Filho, R.B., de Moura Neves, B., Pérez, C.D., de Oliveira, U., Sumida, P., Maranhão, H., Monteiro, L.H., and Carneiro, P. (2022). Neospongodes atlantica, a potential case of an early biological introduction in the Southwestern Atlantic. PeerJ 10, e14347.
Uda, K., Komeda, Y., Fujita, T., Iwasaki, N., Bavestrello, G., Giovine, M., Cattaneo-Vietti, R., and Suzuki, T. (2013). Complete mitochondrial genomes of the Japanese pink coral (Corallium elatius) and the Mediterranean red coral (Corallium rubrum): a reevaluation of the phylogeny of the family Coralliidae based on molecular data. Comp. Biochem. Physiol. Part D Genomics Proteomics 8, 209-219. 10.1016/j.cbd.2013.05.003.
Park, E., Hwang, D.-S., Lee, J.-S., Song, J.-I., Seo, T.-K., and Won, Y.-J. (2012). Estimation of divergence times in cnidarian evolution based on mitochondrial protein-coding genes and the fossil record. Mol. Phylogenet. Evol. 62, 329-345.
There are also some questions to morphology authors are using.
Photos of invasive species is of low resolution. No real comparison with colonies from the type locality. Authors need to take another close-ups illustration sclerites in stalk and in polyps (the last with possibility to see the polyp armature), preferably they have to provide same set of photographs for the species from the type locality.
SEM provided by authors is of low res. there is no good illustration of surface of sclerites. Sclerite sets for each region represented by tiny illustration. The scale for Fig 1A and Fig 1B is not the same - that makes a bad joke - as sclerites at Fig1B are larger. There are also differences in sclerite form and sizes sets for other regions. If they are caused by different scale used or choice of sclerites for illustration - it is hard to tell. Authors need to provide normal-size tables, at least as Supplementary Figures.
Both morphological and genetic identifications need further proof. I would highly recommend authors to consult Dr. Yehuda Benayahu from Tel Aviv University to avoid such mistakes in the future.
there are also some minor problems
1. It is not clear why authors put Dendrophylliidae (Scleractinia) in the keywords.
2. Genus/species names are mostly NOT in Italics (Results but check also other parts of the MS and reference list) - need to be fixed.
3. Introduction section need serious revision. Paragraphs are not logically connected. There are jumping from one subject to another. There is Tubastrea - out of the blue - that is Brazilian invader, but there is no indication where this situation is. It is quite bad for general reader. Apparently authors need to limit it to Lessepsian migration and leave all stray things out.
Authors have to have in mind that we do not abbreviate species in the beginning of the sentence (see line 78).
line 185 "3.2. Molecolar analysis " misprint
There are some statements in the introduction that I do not fully understand. e,g,
lines 80-81 - have in mind that "soft corals' is an accepted term that traditionally encompass octocorals without hard skeleton. so. formally it will be part of Malacalcyonacea and cannot include sea anemones (Hormathiidae)
lines 94-95 "Dendronephthya tends to rapidly populate available substrate, often artificial surfaces, in as little time as two days" - authors mean recruitment or active movement?
lines 95-96 "Dendronephthya is fast growing, and [49] report that the number of Dendronephthya sp. colonies can increase 4-fold in one year following initial recruitment to new habitat" - if I am right that the first part of the sentence is about colony growing (size) and the second is about recruitment of larvae? authors, you cannot mix apples with oranges, need to be re-worded
I have an impression that literature sources are also put randomly. e.g. line 99-100 " In addition to sexual reproduction, Dendronephthya can also reproduce via clonal
propagation [45,46] where autotomized fragments, which are negatively buoyant .." reference 45 is in effect about sexual reproduction.
line 110-111 "In this study, we describe Dendronephthya sp. with an aim to understand the expansion potential of this species at the Mediterranean Sea." - it is claimed by authors that this species is hemprichi?
in lines 116-117 the species is also called Dendronephthya sp.
also my feeling that Conclusions provided by authors also need serious revision.
Round 2
Reviewer 2 Report
It was pointed out IN PREVIOUS ROUND OF THE REVIEW that mitochondrial genes cannot be used for phylogenetic reconstructions within Dendronephthya. The referenceы provided by authors in the cover -letter do not prove utility of genetic markers used by authors to delimit species within Dendronephthya. Apparently both these references were provided by mistake. Williamson et al used 16S to prove their specimens belong to the same species with provision (see page 810) that it is not compelling evidence that their samples represent a single or unique species of Dendronephthya and Figueroa and Baco used order of genes for several genera within Octocorallia.
Again, in revised MS authors added another mitochondrial marker that is not appropriate. The phylogenetic tree (Figure 3), completed by authors shows that genetic markers used by authors are not suited for phylogenetic reconstructions and do not prove identifications or relations between species in the genus Dendronephthya.
Apparently. Authors may try to provided values instead of green circles as values ~75% seems to be rather low.
Table 1. Specimens used for phylogenetic analysis in this study. - need serious revision. there is a mix up between taxa columns ( e.g. genus & family). Authors HAVE TO PROVIDE source of sequence used. It is not "Public" this result was published somewhere, so the source has to be mentioned. It is not clear why species in the table are arranged without order (see e.g. Leptogorgia)
Figure 2 - it is not clear if any of colonies (A-B) were depicted also at close ups (C-D)?
Figure 4. Authors wrote in the cover letter that they provide Supplementary file, but there is no indication in revised MS that this file exist. Anyway. Authors were requested to provide surface of sclerites close-up. This requirement was not fulfilled.
Introduction and discussion sections need further work
Authors are very inaccurate with references used.
eg 43. Authors state at lines 97-99 " Recent phylogenetic analysis using a whole mitochondrial genome provides better resolution of the topology of this family [43]." However only one species of the family was used in [43] and honestly it discuss member of another family. Please re-check all the reference you are using in text.
in the previous round of revision authors were requested to put in full all abbreviated genera that were not used in current section. Please, correct.
In the previous round of revision authors were requested to check latin names - have to be in Italics. Authors, have in mind that species name cannot begin with a Capital (even in the reference list).
Round 3
Reviewer 2 Report
I am OK with the changed title - it may solve some of the problems. However, as the authors do not provide enough evidences - please remove line 26 and modify further text in the Abstract to avoid further ambiguities. there is no suggestion from the genetics. For morphology see comments below.
I have read carefully reply from the authors but there are some questions authors need to solve before paper can be considered for publication.
1. Please provide requested values for fig 3 (phylogenetic tree) at least for Nephtheidae If authors consider that it may result in messy figure, this may be uploaded as a supplementary file.) and discuss these values in the main text
2. Each of collected specimens (fragments) has to have collection/holding number. These numbers have to be provided. these numbers has to be indicated in Material and Methods, in the Table 1 - where sequences are provided. I do not understand lines 149-151 stated "The remainder of the specimens collected were utilized for further morphological examination." If there is no voucher specimen - the paper cannot be published
3. For large spindles (scale 1 mm at figure 4 A) please provide close up to see surface of the spindles. From the figure provided I have an impression that there are differences in surface structure - please show me that it is not so. For figure 4 all sclerited from the Red Sea colonies and all sclerites from the Medditerranian colonies has to came from the same specimen. If they came from several specimens it has to be stated and number has to be indicated for each sclerite.
4. in the section Data Availability Statement authors supposed to cite all products generated during study. Sequences deposited to the GenBank have to be mentioned here.As far as I understand from Figure 3 it has to be 24 sequences deposited.
These 24 sequences have to be mentioned in Table 1 with collection numbers - which sequences came from what specimen. It is mandatory.
5. Table 1
(1) remove column "Genus" - it is not informative
(2) column "NCBI Accession number" is also not informative as under titles rRNA, MSH and COX1 - authors put Accession numbers from MCBI.
(3) about reference authors need to provide publications connected to each accession number. eg. for Dendrophephthya mollis authors are consulting NCBI using Acc number
https://www.ncbi.nlm.nih.gov/nuccore/NC_020456.1
AUTHORS Park,E., Hwang,D.S., Lee,J.S., Song,J.I., Seo,T.K. and Won,Y.J. TITLE Estimation of divergence times in cnidarian evolution based on mitochondrial protein-coding genes and the fossil record JOURNAL Mol. Phylogenet. Evol. 62 (1), 329-345 (2012)
this has to be done for all entries provided
lines 93-94 Pennatuloidea - Superfamily
line 95 - check position of Nephtheidae and change accordingly
https://www.marinespecies.org/aphia.php?p=taxdetails&id=146762
lines 102-103 - have in mind that Dendronephthya IS GENUS. If authors prefere to write in Dendronephthya in general better use Dendronephthya spp. (have in mind if there is end of the sentence only one full stop after spp. or sp. (line 119) "spp." has to be NOT in Italics)
line 134-135 - do we need Fish protocol here?
line 142 - in the remaining MS authors stated that they had 5 colonies... Please provide correct information
line 176 - see comments about sequenced obtained in study. this has to be fixed.
lines 190-196. Abbreviations have to be explained or not used
line 243 - I would advise put Dendronephthya in full - new section
line 261 - genera is plural for genus
line 263. as authors do not prove that this species is hempritchi - please modify this and following 263-276) text of discussion . you can use "cf." but not species name
line line 287 - remove "Family:" (as authors do not use this format throughout the MS)
line 260/ Please indicate Tubastrea as Scleractinia
lines 294-303 - again. no prove about hemprichi.
consider to condense lines 304-335
lines 340-344 - remove 'hemprichi'